# Hydrogel Fluorescence Microsensor with Fluorescence Recovery for Prolonged Stable Temperature Measurements

**DOI:** 10.3390/s19235247

**Published:** 2019-11-29

**Authors:** Hairulazwan Hashim, Hisataka Maruyama, Yusuke Akita, Fumihito Arai

**Affiliations:** 1Department of Micro-Nano Mechanical Science and Engineering, Nagoya University, Furo-cho, Chikusa-ku, Nagoya 464-8603, Japan; hisataka@mech.nagoya-u.ac.jp (H.M.); akita@biorobotics.mech.nagoya-u.ac.jp (Y.A.); arai@mech.nagoya-u.ac.jp (F.A.); 2Faculty of Engineering Technology, Universiti Tun Hussein Onn Malaysia, 86400 Parit Raja, Malaysia

**Keywords:** fluorescence, fluorescent dye, measurement, photobleaching compensation, hydrogel, temperature, sensor

## Abstract

This work describes a hydrogel fluorescence microsensor for prolonged stable temperature measurements. Temperature measurement using microsensors has the potential to provide information about cells, tissues, and the culture environment, with optical measurement using a fluorescent dye being a promising microsensing approach. However, it is challenging to achieve stable measurements over prolonged periods with conventional measurement methods based on the fluorescence intensity of fluorescent dye because the excited fluorescent dye molecules are bleached by the exposure to light. The decrease in fluorescence intensity induced by photobleaching causes measurement errors. In this work, a photobleaching compensation method based on the diffusion of fluorescent dye inside a hydrogel microsensor is proposed. The factors that influence compensation in the hydrogel microsensor system are the interval time between measurements, material, concentration of photo initiator, and the composition of the fluorescence microsensor. These factors were evaluated by comparing a polystyrene fluorescence microsensor and a hydrogel fluorescence microsensor, both with diameters of 20 µm. The hydrogel fluorescence microsensor made from 9% poly (ethylene glycol) diacrylate (PEGDA) 575 and 2% photo initiator showed excellent fluorescence intensity stability after exposure (standard deviation of difference from initial fluorescence after 100 measurement repetitions: within 1%). The effect of microsensor size on the stability of the fluorescence intensity was also evaluated. The hydrogel fluorescence microsensors, with sizes greater than the measurement area determined by the axial resolution of the confocal microscope, showed a small decrease in fluorescence intensity, within 3%, after 900 measurement repetitions. The temperature of deionized water in a microchamber was measured for 5400 s using both a thermopile and the hydrogel fluorescence microsensor. The results showed that the maximum error and standard deviation of error between these two sensors were 0.5 °C and 0.3 °C, respectively, confirming the effectiveness of the proposed method.

## 1. Introduction

Both the measurement and control of the cellular environment with high spatial resolution are essential for investigating cellular conditions systematically, defining normal cell-to-cell variation, quantifying the influence of environmental perturbations, and understanding cellular responses in the tissues and complex environments. For example, the local pH around octacalcium phosphate in granule form was determined using an optical measurement method to analyze the release of inorganic phosphate ions from octacalcium phosphate upon biomolecule absorption [1]. In addition to pH measurement [2], the use of optical sensors to determine various environmental parameters—such as temperature [3,4,5,6] and oxygen concentration [7]—with high spatial resolution is promising for biological investigation.

Optical sensors using fluorescent dyes have significant potential in non-contact measurement for microscale applications. Environmental parameters such as temperature [8,9], oxygen concentration [10,11,12], pH [13,14], and organophosphate compounds [15] can be measured by utilizing fluorescent dyes that are sensitive to these parameters. We have developed fluorescence sensors in which fluorescent dyes are impregnated into various processable materials such as polymers, photoresists, and hydrogels [16]. Fluorescence sensors have paved the way for the investigation of cells, tissues, and the environmental conditions of culture systems, amongst other parameters.

The fluorescence of the sensor can be measured in terms of its fluorescence intensity and fluorescence lifetime [3,7,17,18]. The direct introduction of fluorescent dyes into cells carries the risk of cell damage as well as the efflux of dyes. Therefore, the impregnation of fluorescent dyes into microparticles made of polymers and hydrogels enables low invasivity and long-term measurement. Changes in the fluorescence intensity and fluorescence lifetime of the microsensor can be measured using a microscope with an image sensor, such as a charge-coupled device (CCD). This approach is suitable for multipoint measurement [19]. In addition, positioning the microsensor at arbitrary points is possible through the manipulation of the microsensor using non-contact techniques such as optical tweezers [3,20,21].

When using fluorescent microsensors, photobleaching of the fluorescent dyes that occurs as a result of exposure to the light required for observation is one of the most critical issues [22]. When a sensor is irradiated, some of the fluorescent molecules are excited, resulting in their molecular structure being altered, while the remaining molecules do not emit fluorescence. In microscopic measurement, the effect of photobleaching must be considered because the objective lens focuses the excitation light, and high-energy radiation reaches the fluorescence microsensor. The fluorescence lifetime is used for stability measurements [18,23], however, the equipment required for fluorescence lifetime measurement is expensive and not widely used. In contrast, the equipment for fluorescence intensity measurement can be more easily implemented. In this case, the effect of photobleaching on the evaluation of fluorescence intensity is quite severe, which is related to the concentration of the fluorescent dye.

This work proposes a new method for the prolonged stable measurement of hydrogel fluorescence microsensors based on the measurement of fluorescence intensity. Fluorescence recovery after photobleaching is thought to compensate for the decrease in fluorescence intensity of the microsensor [24,25]. In the hydrogel fluorescence microsensors used in this work, the fluorescent dye is uniformly distributed. When a region of the microsensor is photobleached by the exposure to light, the unbleached fluorescent dye diffuses into the photobleached region. The concentration of fluorescent dye in the photobleached portion is recovered after an appropriate time interval, which enables stable fluorescence measurements to be made over prolonged periods. We fabricated hydrogel fluorescence microsensors and evaluated the time interval of measurement, material, the concentration of photo initiator, and composition of the fluorescence microsensor in order to investigate the effect of these parameters on photobleaching. In addition, the effect of the microsensor size on the stability of the fluorescence intensity was also evaluated. A temperature-sensitive fluorescent dye was used for temperature measurement inside a stage-top chamber, and a thermocouple was used for calibration to confirm the effectiveness of the proposed measurement method.

## 2. Materials and Methods

### 2.1. Principle of Photobleaching Compensation for the Hydrogel Fluorescence Microsensor

Fluorescence intensity is represented by Equation (1) [26].
(1)$$I=I_{0}\Phi \varepsilon Cxexp(-t/\tau )$$
where I [W] is the fluorescence intensity emitted from the fluorescent material per unit time, $I_{0}$ [W] is the excitation light flux of the fluorescent material, Φ is the fluorescence quantum yield, C [mol/L] is the concentration of the fluorescent dye, ε is the absorbance index [L/mol·cm], x [cm] is the diameter of the hydrogel fluorescence microsensor, t [s] is the total exposure time, and τ [s] is the decay coefficient of the fluorescence intensity. The structure of some of the fluorescent dye molecules changes to a non-fluorescent structure with every exposure to the excitation light. Therefore, the concentration of fluorescent dye and the fluorescence intensity decrease with increasing measurement repetition. This phenomenon, known as photobleaching, is one of the most serious challenges in achieving long-term stable measurement based on fluorescence intensity. The recovery of the concentration of the fluorescent dye after exposure is a promising approach for addressing this limitation.

Figure 1 shows a schematic diagram of fluorescence intensity recovery in the hydrogel fluorescence microsensor. The microsensor is made of photo-crosslinkable hydrogel, water, and fluorescent dye. The fluorescent dye is uniformly distributed inside the microsensor, which is made of the photo-crosslinkable hydrogel. In this method, a confocal laser scanning microscope is used to measure the fluorescence. The excitation laser is scanned inside the microsensor to measure fluorescence. All fluorescent dyes in the path of the excitation laser will photo-bleach by laser excitation. The dependency of photo-bleaching to the intensity of excitation laser has been reported previously in reference [27]. The study indicated that the higher the excitation laser intensity, the higher the photobleaching of fluorescent dye. Distribution of laser intensity is determined by the laser profile of the focused laser beam. Laser intensity of the laser focused area is considered higher than that of the other part. Therefore, based on reference [27], the effect of photo-bleaching at the laser focused area is considered higher than that at the other part of total excited area. The thickness of the laser focused area is highlighted in yellow in Figure 1a. Fluorescent dye in the microsensor emits fluorescence and some of the fluorescent dye molecules are bleached. As a result, the concentration of unbleached fluorescent dye decreases as shown in Figure 1b. The bleached and unbleached fluorescent dyes then diffuse inside the hydrogel fluorescence microsensor to give uniform concentrations of both dyes across the microsensor. After diffusion of the dyes is complete, the concentration of unbleached fluorescent dye at the laser focused area is recovered, as shown in Figure 1c.

The time required for the diffusion of unbleached fluorescent dye depends on the concentration of the hydrogel. The low-concentration hydrogel allowed the fluorescent dye to demonstrate high diffusivity. The relationship between the size of the hydrogel fluorescence microsensor and the thickness of the excitation area is critical for the long-term stability of the fluorescence intensity. In photobleaching compensation, unbleached fluorescent dye moves to the excitation as some of the fluorescent dye molecules in the excitation area are bleached by irradiation with the excitation light. The thickness of the excitation area D is calculated using Equation (2) [25].
(2)$$D=\frac{0.67\lambda_{ex}}{n-\sqrt{n^{2}-{\left(NA\right)}^{2}}}$$
where $\lambda_{ex}$ is the wavelength of the excitation light, n is the refractive index of the solution, and NA is the numerical aperture. When $\lambda_{ex}$, n, and NA are 561 nm, 1.33 (water), and 0.45, respectively, D is calculated as approximately 4.7 µm. The amount of unbleached fluorescent dye is determined from the concentration of fluorescent dye and the sensor size. Therefore, we assume that a larger sensor represents a greater resistance to photobleaching.

In summary, using a hydrogel fluorescence microsensor with a size that is larger than D and an appropriate interval time that is longer than the diffusion time of the unbleached fluorescent dye should result in long-term measurement stability.

### 2.2. Experimental System

Figure 2 shows a schematic diagram of the experimental system. A commercial inverted microscope (ECLIPSE Ti-E, Nikon Co., Tokyo, Japan) with confocal laser scanning unit (CSU-X1, Yokogawa Electric Co., Tokyo, Japan), objective lens (CFI Apochromat 20 XC, Nikon Co., Tokyo, Japan), and electron multiplying-charge coupled device (iXon Ultra, Andor Technology Ltd., Belfast, UK) were used to acquire the confocal fluorescence images. The diameter of the pinhole of CSU-X1 was 50 µm. The numerical aperture (NA) of the objective lens was 0.45. During the fluorescence measurement, the focus of the fluorescence image was maintained using the automatic focusing system (Perfect Focus System) of the Ti-E. An excitation laser with a wavelength of 561 nm was used to excite the fluorescent dye in the hydrogel fluorescence microsensor. A stage top incubator with a feedback temperature controller (ZILCS, Tokai Hit Co. Ltd., Shizuoka, Japan) was placed on the microscope stage. The accuracy of the temperature control was ±0.3 °C. The fluorescence microsensors were attached to the glass surface of the glass bottom dish. First, 100 µL of the solution containing the fluorescence microsensors was dropped onto the glass surface. The fluorescence microsensors were fixed on the glass surface by the evaporation of the solution as a result of heating for 30 min at 80 °C on a hot plate. After fixing the fluorescence microsensors, 2 mL of deionized water was introduced into the glass bottom dish. The dish was then placed in the stage top incubator. A thermopile was used to measure the temperature of the deionized water inside the glass bottom dish. The acquisition of fluorescence images and analysis of the fluorescence intensity were performed using NIS-Elements Ar software (Nikon Co., Tokyo, Japan).

### 2.3. Fabrication of the Hydrogel Fluorescence Microsensor

Hydrogel and polystyrene fluorescence microsensors were fabricated to demonstrate the effectiveness of photobleaching compensation as a result of the diffusion of unbleached fluorescent dye inside the hydrogel fluorescence microsensor. Appendix A shows the fabrication processes for the hydrogel and polystyrene fluorescence microsensors. Both fluorescence microsensors were impregnated with Rhodamine B (Wako Pure Chemical Corp., Osaka, Japan), which is a temperature-sensitive fluorescent dye. The excitation and emission wavelength of Rhodamine B are 561 nm and 580 nm, respectively [28].

The hydrogel fluorescence microsensor consisted of PEGDA 575 (Sigma-Aldrich), deionized water, and Omnirad 1173 (BASF Japan Ltd., Tokyo, Japan), which is a photo initiator. The hydrogel fluorescence microsensor was fabricated using different concentrations of PEGDA 575 (9% and 99%) and Omnirad 1173 (0.5%, 1.0%, 1.5%, 2.0%, 2.5%, and 3.0%) to evaluate the effects of the photo-crosslinkable hydrogel and photo initiator concentrations.

The fabrication process for the hydrogel fluorescence microsensor (9% PEGDA575, 1.0% Omnirad 1173) was as follows (Appendix A).
First, 100 µL PEGDA 575, 1000 µL deionized water, 100 µL Rhodamine B aqueous solution (1 g/L), and 10 µL Omnirad 1173 were mixed, and 100 µL of the mixture was introduced into 1 mL of mineral oil in a 1.5 mL microtube.The hydrogel fluorescence microsensor solution was emulsified by stirring.The hydrogel fluorescence microsensor emulsion was photo-polymerized by ultraviolet light illumination (wavelength: 330–380 nm) for 10 min with Xcite-LED1 (Excelitas Technologies Corp., Waltham, MA, USA).The polymerized microsensors were centrifuged at a speed of 14,000 *g* for 15 min to concentrate the hydrogel fluorescence microsensors at the bottom of the microtube.The hydrogel fluorescence microsensors were washed by replacing the mineral oil with deionized water.

A polystyrene fluorescence microsensor was fabricated for comparison with the hydrogel fluorescence microsensor as fluorescent dye does not diffuse in polystyrene. The polystyrene fluorescence microsensor consisted of 20 µm polystyrene microparticles (Duke Scientific Corp., Palo Alto, CA, USA) and Rhodamine B. The fabrication process is described in Appendix A [28].
First, 1 µL of polystyrene microparticles was stained with Rhodamine B ethanol solution (1 g/L).The stained fluorescence microsensors were centrifuged at a speed of 14,000 *g* for 15 min.The stained microsensors were washed three times using DI water.

Figure 3 shows the fluorescence images of the hydrogel fluorescence microsensor and polystyrene fluorescence microsensor. The size of the fabricated hydrogel fluorescence microsensors ranged from a few to 60 µm as shown in Appendix A. The evaluation of the photobleaching compensation was performed using 20 µm hydrogel fluorescence microsensors.

## 3. Results

### 3.1. Comparison of Photobleaching Compensation for Different Sensor Materials

Figure 4 shows the comparison of the fluorescence intensity recovery for the different microsensor materials. To compare the diffusivity of the fluorescent dye in the microsensors, the concentrations of hydrogel material (PEGDA 575) used to form the hydrogel fluorescence microsensors were set at 9% and 99%. The concentration of Omnirad 1173 was 3.0%. The size of both hydrogel fluorescence microsensors was 20 µm. The exposure and interval times for excitation were 1 s and 10 min, respectively. The power of the excitation laser was 20 mW. The temperature was maintained at 25 °C. The hydrogel fluorescence microsensor with 9% PEGDA 575 showed sufficient recovery of fluorescence intensity for stable fluorescence measurement. The standard deviation of the difference from the initial fluorescence intensity was within 1%. In contrast, the hydrogel fluorescence microsensors with 99% PEGDA 575 and the fluorescent polystyrene exhibited a continuous decrease in fluorescence intensity (99% PEGDA: approximately 10%, polystyrene: approximately 20%). The reason both the 99% PEGDA 575 and polystyrene fluorescence microsensors did not show recovery of their fluorescence intensity is thought to be that the unbleached fluorescent dye was not able to diffuse effectively into the depleted region within the interval time. These results indicate that the diffusivity of the fluorescent dye inside the microsensor was one of the critical parameters for photobleaching compensation.

### 3.2. Effect of Measurement Interval Time on Photobleaching Compensation

The fluorescence intensity of 20 µm hydrogel fluorescence sensors (9% PEGDA 575, 3.0% Omnirad 1173) was repeatedly measured at various time intervals to investigate the required interval time for the recovery of fluorescence intensity at each measurement. The exposure time was set at 1 s. The interval time was changed to 0, 1, 2, 4, 6, and 8 s. An excitation laser with wavelength of 561 nm was set at 20 mW. The temperature was maintained at 25 °C throughout the experiments. Experiments with each of the interval periods were repeated 100 times. Figure 5 shows the relationship between the variation of fluorescence intensity of hydrogel fluorescence microsensors with different interval times. The relative fluorescence intensity of the hydrogel fluorescence microsensors decreased by more than 1% for interval times shorter than 4 s. However, for interval times of 6 or 8 s, the fluorescence intensity decreased slightly less than 1% after 100 excitations. These results showed that photobleaching compensation depends on the interval time between exposures.

### 3.3. Effect of Photo Initiator Concentration on Photobleaching Compensation

Figure 6a shows the relationship between the variation of fluorescence intensity and the concentration of photo initiator. For this investigation the excitation and interval times were set at 1 s and 6 s, respectively. The temperature was maintained 25 °C throughout the experiments. Each point in the figure shows the average value for 10 hydrogel fluorescence microsensors with different concentrations of photo initiator. Hydrogel fluorescence microsensors with less than 1.5% photo initiator did not recover well. In contrast, hydrogel fluorescence microsensors with more than 2% photo initiator exhibited clear recovery from photobleaching (Figure 6b). The effect of the concentration of photo initiator on the diffusion coefficient of Rhodamine B in PEGDA 575 hydrogel has been reported previously [29]. The findings of the study indicated that the higher the concentration of photo initiator, the higher the diffusion coefficient of Rhodamine B in the PEGDA 575 hydrogel. Based on reference [29], a concentration of photo initiator is considered influential for the recovery of fluorescence in the fluorescence microsensor.

### 3.4. Effect of Sensor Size on Fluorescence Intensity Recovery

Figure 7 shows the relationship between the relative fluorescence intensity recovery and the size of the hydrogel fluorescence microsensor with 9% PEGDA 575. The concentration of Omnirad 1173 was 3.0%. The diameter of the hydrogel fluorescence microsensors investigated was 5, 10, 20, 30, 40, and 50 µm. The exposure time and interval time were set at 1 s and 9 s, respectively. The temperature was maintained at 25 °C throughout the experiments. Each measurement was repeated 900 times. Each point shows the averaged value and standard deviation of the relative fluorescence intensity of 5 points before and after every 100 repetitions. The decrease of fluorescence intensity after 900 repetitions for the 5 µm sensors was approximately 5%. While, the decreases in fluorescence intensity after 900 repetitions for the 10, 20, 30, 40, and 50 µm sensors were less than 3%.

The excitation area *D* for our experimental setup was calculated to be 4.7 µm using Equation (2). The fluorescent dye in the excitation area becomes non-fluorescent on irradiation. It is thought that in the proposed method the unbleached dye in the unexcited area diffuses into the excitation area, and the concentration of fluorescent dye recovers. Therefore, fluorescence microsensors larger than *D* (10, 20, 30, 40, and 50 µm in diameter) showed better recovery properties than the 5 µm fluorescence microsensor (similar size to *D*). Fluorescence microsensors larger than D exhibited a decrease in fluorescence intensity within 1% until 400 repetitions.

### 3.5. Temperature Measurement Using Hydrogel Fluorescence Microsensors

To demonstrate prolonged stable temperature measurement with photobleaching compensation, calibration of the hydrogel fluorescence microsensor was performed in the incubator at controlled temperatures from 36 °C to 40 °C in 0.5 °C steps. The hydrogel fluorescence microsensors were attached to the bottom of a glass bottom dish. The fluorescence intensities of the microsensors were then measured at each temperature in the glass bottom dish filled with 2 mL of deionized water. The temperature of the deionized water was monitored by thermopile and controlled automatically. Appendix A shows the calibration result of the relative fluorescence intensity based on the fluorescence intensity at 36 °C. The horizontal axis shows the temperature measured by the thermopile and the vertical axis shows the relative fluorescence intensity based on the fluorescence intensity at 36 °C. The relative fluorescence intensity of the hydrogel fluorescence microsensor showed a continual decrease with increasing temperature. The temperature sensitivity of the hydrogel fluorescence microsensor was calculated using the least-squares method. From these calibration results and linear fitting, the temperature sensitivity and standard deviation of temperature measurement were calculated to be −1.4%/°C and ± 0.5 °C, respectively, as shown in Equation (3) [20].
(3)$$\frac{I}{I_{0}}=-0.014Temp+1.5$$

Figure 8 shows the experimental result for temperature measurement in the deionized water in the glass bottom dish during temperature variation. The horizontal axis shows the elapsed time, while the vertical axis shows the temperature measured using the hydrogel fluorescence microsensor and the thermopile. The exposure and interval times were set to 1 s and 19 s, respectively. The total number of measurement repetitions using the hydrogel fluorescence measurement was 270, and the duration of the measurements was 5400 s. The temperature of the water in the glass bottom dish was varied from 36 °C to 40 °C using the temperature controller. In Figure 8, the black points represent the temperature measured by the thermopile. The red points represent the temperature measured using the hydrogel fluorescence microsensor. The results show that the measured temperature values of the hydrogel fluorescence microsensor and thermopile exhibited good agreement. The maximum difference and standard deviation of the difference of the measured temperature between the sensors were 0.8 °C and 0.3 °C, respectively. These findings confirmed the effectiveness of photobleaching compensation for stable long-term fluorescence measurement.

## 4. Discussion

In photobleaching compensation of hydrogel fluorescence microsensors, the diffusivity of the fluorescent dye inside the hydrogel, the time for diffusion of the fluorescent dye, and the stability of the fluorescence intensity are critical parameters. The diffusivity of fluorescent dyes is affected by the materials and composition of the microsensors. Rhodamine B was unable to diffuse inside the polystyrene microbeads in deionized water since the density of the crosslinks in the microbeads is high. As a result, no fluorescence recovery was observed for the polystyrene microbeads. For the hydrogel fluorescence microsensors, the difference in the concentration of the hydrogel affects the diffusion of the fluorescent dye. For the high hydrogel concentration (99% PEGDA 575), the crosslinking density of the hydrogel is high, and the mobility of the fluorescent dye is low. As a result, fluorescence recovery was not observed. In contrast, at a low hydrogel concentration (9% PEGDA 575), the main component of the interior of the hydrogel microsensor is deionized water, and the mobility of the fluorescent dye is high. As a result, fluorescence recovery was clearly observed. In addition to PEGDA 575, other hydrogel materials should be investigated and the longevity of the fluorescence microsensors in solutions such as culture medium should be evaluated as part of is future work.

The hydrogel fluorescence sensor with 9% PEGDA 575 was found not to be effective for fluorescence recovery with an interval time of 6 s when the concentration of the photo initiator was 1.5% or less. In contrast, the effectiveness of the fluorescence recovery with a 6 s interval time was confirmed at concentrations of 2.0–3.0%. The effect of photo initiator concentration on the diffusion properties of fluorescent dyes—other than Rhodamine B—in hydrogel materials will be considered in future work.

The time required for the diffusion of the fluorescent dye inside the hydrogel fluorescence microsensor depends on both the amount of photobleached fluorescent dye and the diffusion rate of the fluorescent dye determined by Fick’s law. Photobleaching occurs as a result of the molecular structure change of the fluorescent dye from a fluorescent molecule to a non-fluorescent molecule due to the transfer of additional excitation energy to the unstable fluorescent dye during excitation. Therefore, the longer the excitation time in a single exposure is, the greater the photobleaching that occurs. To reduce photobleaching, it is useful to shorten the exposure time and reduce the fluorescence intensity. Increasing the concentration of the fluorescent dye increases the fluorescence intensity that can be achieved with short exposure times, based on the relationship in Equation (1). Evaluation of the influence of the exposure time, fluorescence intensity, and initial concentration of fluorescent dye will form part of our future work.

The sensitivity of fluorescence measurements of environmental parameters depends on the environmental sensitivity of the quantum yield in each fluorescence indicator. In the case of Rhodamine B, the relationship between the relative fluorescence intensity and the temperature was calibrated as −1.5%/°C, as shown in Appendix A. If the fluorescence intensity decreases at 1% at a constant temperature, the 0.67 °C temperature increase is measured by the fluorescence microsensor. This error value is similar to the precision of temperature measurement (0.5 °C). The measurement error due to the variation of fluorescence intensity is therefore considered a severe limitation for detailed biological investigations.

In addition, evaluation of photobleaching compensation using fluorescent dyes other than Rhodamine B is needed to validate the generality of the proposed method. Appendix A show the relative fluorescence intensity of hydrogel fluorescence microsensors with an incorporated pH-sensitive fluorochrome, FITC, and an oxygen concentration sensitive fluorochrome, Tris(2,2′-bipyridyl)dichlororuthenium(II) hexahydrate (Ru(bpy)_2_Cl_2_) [28,30]. The hydrogel fluorescence microsensors containing FITC and (Ru(bpy)_2_Cl_2_) both showed fluorescence recovery from photobleaching. These results indicate that photobleaching compensation using a low-concentration hydrogel microsensor can be applied to various fluorescent dyes for measurements in microscale environments. The long-term stable measurement of various physiological properties of cells and tissues using hydrogel fluorescence microsensors will be part of our future work.

## 5. Conclusions

In this work, a hydrogel fluorescence microsensor for prolonged stable temperature measurement was proposed, and the conditions for the recovery of the fluorescence intensity were evaluated. The experiments confirmed photobleaching compensation at 9% PEGDA 575. The required interval time for compensation of the hydrogel fluorescence microsensor assembled from 9% PEGDA 575 was 6 s. In addition, the degree of polymerization of the hydrogel affected the diffusivity of the fluorescent dye inside the hydrogel fluorescence microsensor. A photo initiator concentration of 2.0% or more was effective for photobleaching compensation. We confirmed that the stability of the fluorescence intensity depends on the size of the microsensor.

Based on these results, temperature measurement of deionized water in a glass bottom dish in a stage-top incubator was performed using a thermocouple and hydrogel fluorescence microsensor. In repetitive measurements (total repetition: 270 times, total duration: 5400 s), the maximum and standard deviation of measurement error between the thermocouple and hydrogel fluorescence microsensor were 0.8 °C and 0.3 °C, respectively, which confirmed the effectiveness of the hydrogel fluorescence microsensors for long-term measurement of physiological parameter.

Arbitrary patterns of hydrogel fluorescence microsensors can be made on glass or polymer surfaces using photolithography. Physiological measurements of cells, tissues, and the culture environment will form part of our future work. The proposed method will make a significant contribution to the analysis of cellular responses during culture and differentiation using non-contact measurement of the cellular environment.

## Figures and Tables

**Figure 1 sensors-19-05247-f001:**
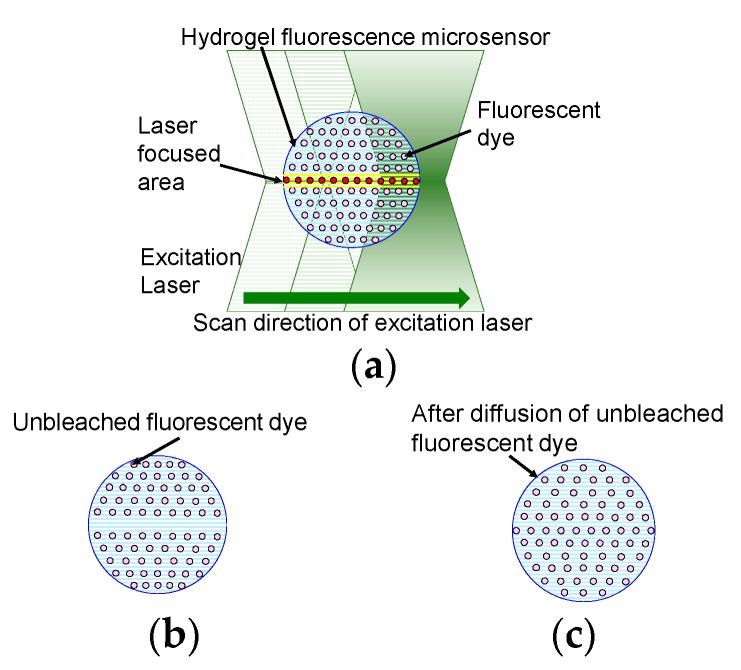
Schematic diagram of fluorescence recovery after photobleaching in a hydrogel fluorescence microsensor. (**a**) Scanning of the hydrogel fluorescence microsensor with the excitation laser for measurement; (**b**) Diffusion of unbleached fluorescent dye in the hydrogel fluorescence microsensor; (**c**) Hydrogel fluorescence microsensor with recovered unbleached fluorescent dye distribution.

**Figure 2 sensors-19-05247-f002:**
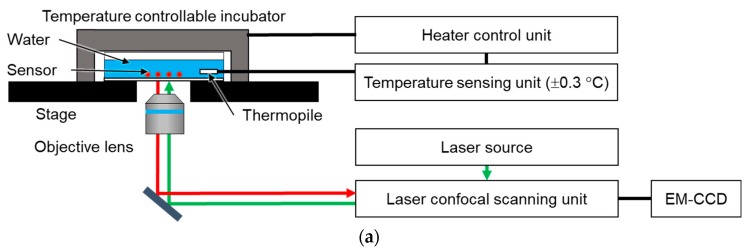
Experimental system. (**a**) Schematic diagram of the experimental system; (**b**) Photograph of the stage-top incubator; (**c**) Temperature control unit for the stage-top incubator.

**Figure 3 sensors-19-05247-f003:**
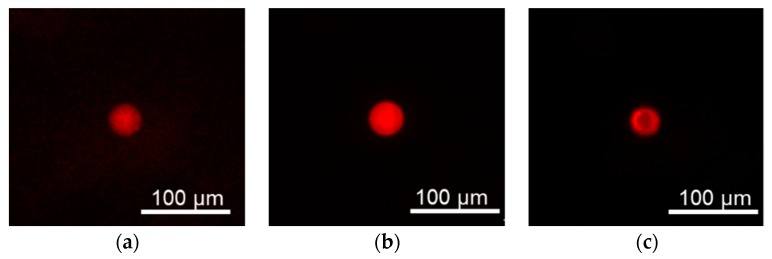
Fluorescence images of fabricated microsensors. (**a**) Hydrogel fluorescence microsensor (low-concentration; 9% PEGDA 575); (**b**) Hydrogel fluorescence microsensor (high-concentration; 99% PEGDA 575); (**c**) Polystyrene fluorescence microsensor.

**Figure 4 sensors-19-05247-f004:**
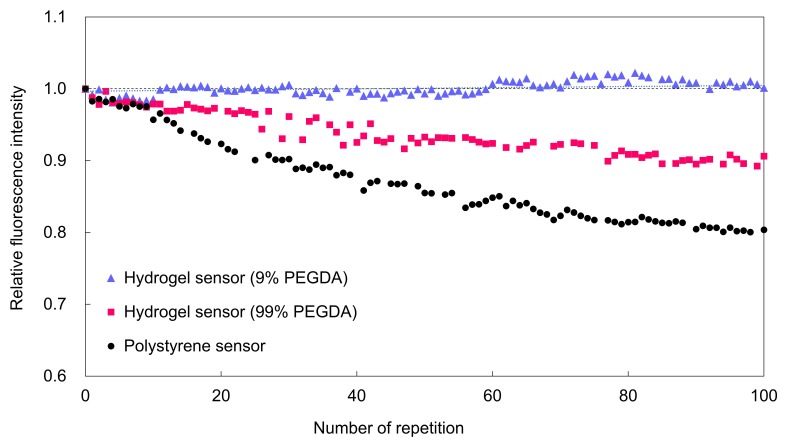
Comparison of the fluorescence intensity recovery for the different sensor materials.

**Figure 5 sensors-19-05247-f005:**
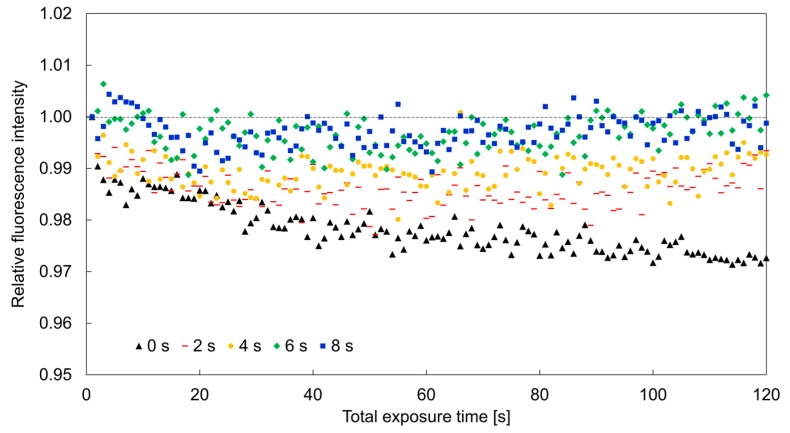
Relationship between the variation of fluorescence intensity of the hydrogel fluorescence microsensor (9% PEGDA 575) and the interval time between exposures.

**Figure 6 sensors-19-05247-f006:**
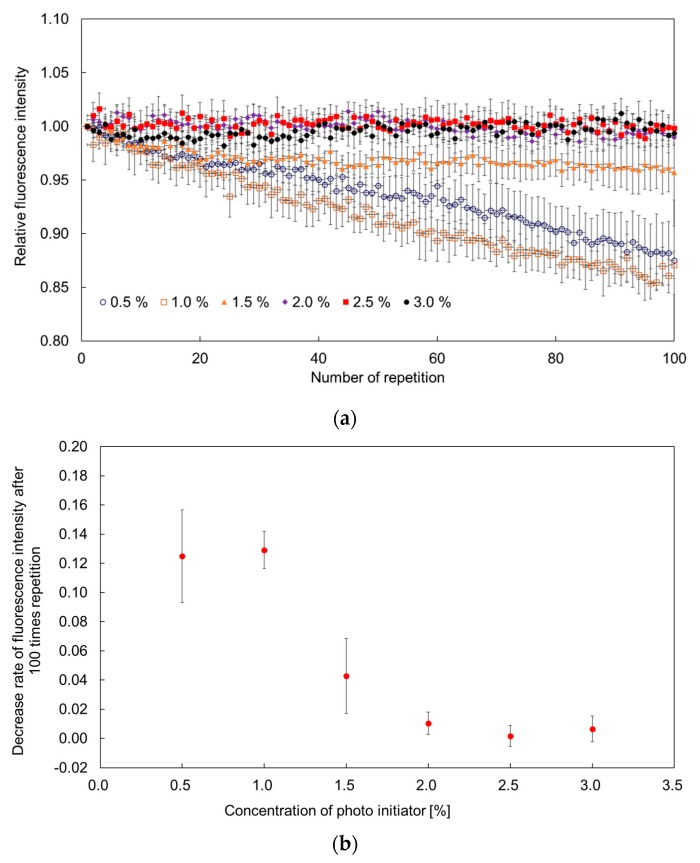
Evaluation of the effect of photo initiator concentration on photobleaching compensation, (**a**) Comparison of the relative fluorescence intensity variation of hydrogel fluorescence microsensors with various concentrations of photo initiator; (**b**) Relationship between the photo initiator concentration and the fluorescence intensity decrease rate after 100 measurements.

**Figure 7 sensors-19-05247-f007:**
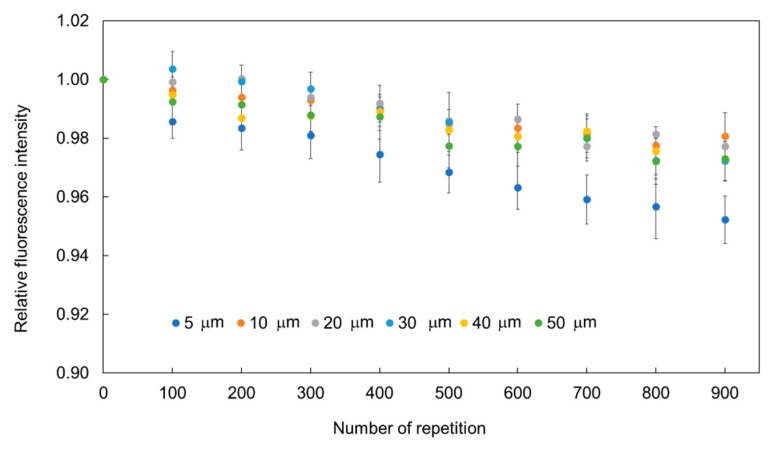
Relationship between the recovery of relative fluorescence intensity and microsensor size.

**Figure 8 sensors-19-05247-f008:**
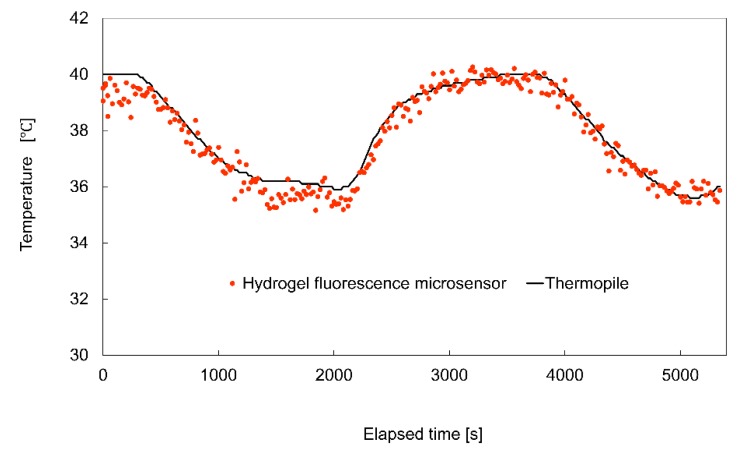
Comparison of the temperatures measured by the hydrogel fluorescence microsensor with photobleaching compensation and a thermopile.

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
