# Peer review of "Hydrogel Fluorescence Microsensor with Fluorescence Recovery for Prolonged Stable Temperature Measurements"

_sensors, 2019, doi:10.3390/s19235247_

Round 1

Reviewer 1 Report

The authors describe a technique to evaluate fluorescence recovery of photosensitive dyes after photobleaching in PEG-based hydrogels and in polystyrene.

The collected data may be to some extent useful for the scientific community, and they are collected in agreement with scientific methods, but a number of severe flaws and inaccuracies prevent the article from being published in the present form. 

In particular, I underline the following aspects:

Overall consideration: The language used is very often of low quality, and sometimes it prevents the reader from understanding what the authors are trying to explain. Moreover, sentences are sometimes unspecific (e.g.: "The proposed method will contribute to bioengineering and biology" (last sentence of the paper)). Some expressions are not appropriate for a scientific journal (e.g. "high magnification objective" for a 20x lens and "perfect focusing system" (page 4, lines 140-143)). It is not clear how the microsensors look like in the experimental setup. How are these implemented in the water chamber depicted in figure 2a? How are the hydrogel particles kept in focus during the experiments? For temperature measurements (paragraph 3.5), it is stated that "microsensors were fixed on the bottom of the glass-bottom dish" (page 9, line 260). This explanation is inaccurate.  The first part of the introduction (lines 41-50) is too general and does not relate with the following of the article: it is not clear how the fluorescence sensor could help biological investigations named in citations 1-6. References for the equations (1)-(3) are missing. The concentration of PEGDA575 in 99% hydrogel is very high. Do the authors polymerize the hydrogel via UV-irradiation for the same time (i.e. 10min) they use to polymerize the 9% hydrogel? If so, to the best of my knowledge, swelling capabilities are reduced and almost no water absorption can be induced. However, differences in relative fluorescence intensity are less than 10% with respect to the 9%-sensor (fig. 5). Authors could have commented on that. Figure 6 shows a huge amount of data which are difficult to read. Moreover, no fitting is proposed, which would have improved the quantification of the difference in fluorescence intensity between the different interval times. Most importantly, difference stated by the authors are in this case on the order of 1%. Is their technique so sensitive to detect in a statistically significant way such small differences? Comments on that would have been appreciated and would have improved the soundness of their technique. Meaning of error bars in figure 7 and 8 is not specified: is it standard deviation?  Paragraph 3.4 states that fluorescence recovery depends on the size of microsensors: however, only 5µm-sensors seem to show a trend different from the other dimensions (dots representing 10,20,30,40 and 50µm sensors are mostly overlapping in fig.9). In absence of any other statistical consideration, this conclusion (page 9, lines 253-254) is very weak. Figure organization is unusual for a scientific journal: authors should have preferred to reduce the number of figures to max.4-5 (not 13), possibly by incorporating more graphs in one figure and using letters to distinguish between the graphs (e.g. figures 7 and 8 could have been merged in one figure (with graph 7a and graph 7b), and fig. 12 and 13 in one single figure 12 a-d).  A number of typos (e.g. "hydrogen" in caption of fig.11) could have been easily avoided by re-reading the text carefully. Authors are advised to do so at their next submission.

Reviewer 2 Report

The authors propose a method using hydrogels as fluorescent microsensors by encapsulating Rhodamine B for fluorescence intensity measurements. The work is interesting and could potentially have future applications. I have a few suggestions and questions to the authors.

(1) what is the size distribution of the microsensors produced? What is the longevity of these microsensors in water (retention of size)? For potential applications in living cells, the current size of the microsensors are too big for typical cell size. Could the authors comment on this?

(2) The authors claim that unpolymerized precursor solution remain inside the sensors. Do the authors have some background information, additional measurements or reference they could cite to support their claims? 

(3) Could the authors add some references of fabricated microsensors 

line 99: inflorescence

trough -> i think the authors mean throughout

Reviewer 3 Report

This paper put forward an interesting strategy. It took use of the fluorescence recovery of dyes in hydrogel materials to overcome the optical quenching problem, with which long-term and stable temperature determination became possible with such hydrogel microsensors. The results are interesting. However, there might be some problem with the mechanism raised by the authors and therefore major revision is suggested before possible publication of this paper.

As in Figure 1 and Page 3, the authors said that “the fluorescence dye at focal plane” was “partly bleached by the excitation laser”, and therefore, the other dyes would move in to recover the fluorescence under observation for temperature determination.

However, not only the dyes on the focal plane, but all the dyes in the light path might be photo-bleached, although the out-of-focus light is blocked by the pinhole during image acquirement. This might be a huge area/volume.

As a result, in this system, the FRAP process would be influenced by not only the spatial diffusion, but also the vertical diffusion of the dyes (which would be affected by the thickness of the as-fabricated microsensor and the size of the pinhole).

Quantitative or semi-quantitative analysis of the FRAP behavior (e.g. velocity) of dye would be helpful, especially when the microsensor has different size and thickness.

And also, the words in Figure 1 should not be “unbeached dye” but “unbleached dye”.

Moreover, it seems that too many images were included in the manuscript and some of them might be removed to the ESI.

Round 2

Reviewer 1 Report

The authors have significantly improved the overall quality of the manuscript. Their work is now clear and the results understandable. Advantages and limitations of their technique are properly addressed.

Therefore, I now recommend the article for publication in sensors.

Please note that Fig.3 is messed up (drawings indicated with letter (a) refer to (b) and vice versa). Please amend this prior to publication.

Reviewer 3 Report

I’m very sorry to see that the questions I raised have not been resolved.

Last time I mentioned that, there are “some problem with the mechanism raised by the authors”. There are serious mistakes in the descriptions on Page 3 and in Figure 1 which are closely related with the kernel of this technique (with the reasons I explained previously). However, the authors did not make any revision at all.

I mentioned (so did the first referee) “too many images were included in the manuscript and some of them might be removed to the ESI”. Unfortunately, the authors just “merged four original figures into two figures” without more revision.
